# Acute coronary syndrome versus acute myocarditis in young adults–value of speckle tracking echocardiography

**Paulina Wieczorkiewicz** ⬤ *, **Karolina Supel** ⬤, **Katarzyna Przybylak, Michal Kacprzak, Marzenna Zielinska**

The Department of Interventional Cardiology, Medical University of Lodz, Lodz, Poland

* paulina.wieczorkiewicz@umed.lodz.pl

**Data Availability Statement:** All relevant data are within the manuscript and its Supporting Information files.

## Abstract

### Purpose

Comparing myocarditis with an acute coronary syndrome (ACS)-like presentation and acute myocardial infarction (AMI) poses an important clinical challenge. The purpose of the study was to investigate the diagnostic value of the clinical, laboratory and especially echocardiographic characteristics including speckle tracking echocardiography (STE) of patients with ACS-like myocarditis and AMI.

### Methods

We conducted a retrospective analysis comparing 69 symptomatic patients ($\leq$ 45 years old), hospitalized at the Department of Interventional Cardiology (Medical University of Lodz, Poland) between April 2014 and June 2021 with an initial diagnosis of ST-segment elevation myocardial infarction.

### Results

37 patients with the cardiac magnetic resonance–confirmed acute myocarditis and 32 patients diagnosed with AMI based on the clinical presentation, electrocardiogram and the presence of a culprit lesion on the coronary angiography were analysed including echocardiography parameters. On STE analysis an average global longitudinal (GLS), radial and circumferential strain including three—layers observation were significantly lower (absolute value) in patients with AMI versus acute myocarditis (p<0.05). There was no significant difference in Endo/Epi ratio (p = 0.144) between the groups. An average GLS < (-17.5) represented the optimal cut-off value for the myocarditis diagnosis.

### Conclusion

In patients with AMI a significant reduction of global and three-layers strains compared to patients with myocarditis was detected. Furthermore, our analysis also confirmed the discriminative pattern of myocardial injury between the groups.

**Funding:** The authors received no specific funding for this work.

**Competing interests:** The authors have declared that no competing interests exist.

## Introduction

Myocarditis is an inflammatory heart disease induced by both infectious and non-infectious causes [1–3]. It is estimated that the incidence of myocarditis fluctuates between 0.02 and 1.5% of the general population with the average age of patients from 20–51 years, mostly men [1, 4, 5]. Due to the heterogeneity of clinical presentation, which can range from asymptomatic to acute coronary syndrome (ACS) -like and acute heart failure, the diagnosis of acute myocarditis (AM) is still challenging [6–9]. When myocarditis is presented with chest pain, electrocardiographic changes with ST-segment deviation, raised cardiac troponin levels, differentiation from myocardial infarction (MI) is necessary [9–11]. Echocardiography findings including global ventricular dysfunction, regional wall motion abnormalities or diastolic dysfunction are also non-specific [6, 10]. Thus, a coronary angiography has been still performed in patients with inflammatory heart diseases in order to exclude obstructive coronary artery disease (CAD) [8, 10]. A final diagnosis of myocarditis is possible only on endomyocardial biopsy (EMB), which is still the diagnostic gold standard but due to its invasive character, low availability and small numbers of multidisciplinary teams with complementary competences in EMB procedure, extremely limited in clinical practice [6, 7, 10, 12–14]. Hence, cardiac magnetic resonance imaging (CMR) has recently emerged as a reference standard for the diagnosis of acute myocarditis due to its ability to identify myocardial oedema and inflammation, as well as scarring associated with MI [15–17]. It seems to be reasonable to first use CMR as a noninvasive tool for the tissue characterization of the myocardium in clinically stable patients presented with chest pain, ST segment elevation in electrocardiogram (ECG) and normal coronary arteries [6, 18]. However, CMR is expensive and not always available in many areas worldwide [6, 8, 10, 19]. One of the most commonly used devices in most centres is conventional transthoracic echocardiogram (TTE) but is often not sufficiently sensitive to detect subtle myocardial dysfunction [7, 8]. Therefore, other recently developed echocardiography-based methods such as speckle tracking echocardiography (STE) imaging emerged as an area of interest in the differential diagnosis of patients with ST-segment elevation [7, 10]. Two-dimensional STE allows clinicians more sophisticated assessment of left ventricular (LV) systolic and diastolic function including three-layers analysis [20].

The aim of this research is to investigate the diagnostic value of global longitudinal (GLS), radial (GRS) and circumferential strain (GCS) imaging by echocardiography as a differentiating tool in patients hospitalized with an initial diagnosis of ST-segment elevation myocardial infarction (STEMI). The following article was presented in accordance with the STROBE reporting checklist.

## Methods

### Study population

We retrospectively analysed 69 symptomatic patients (≤45 years old) urgently admitted to the Department of Interventional Cardiology (Medical University of Lodz, Poland) within 12h of onset symptoms between April 2014 and June 2021 with an initial diagnosis of STEMI. ST segment elevation was defined as a J point elevation in two contiguous leads with the cut-points: ≥0.25 mV in men below the age of 40 years, ≥0.2 mV in men over the age of 40 years, ≥0.15 mV in women in leads V2 –V3 and/or ≥0.1 mV in other leads. In order to confirm or exclude obstructive CAD, the invasive coronary angiography was performed in all cases. Patients were classified into two groups, according to their final diagnosis. Group 1: Patients diagnosed with MI based on the clinical presentation, ECG and the presence of a culprit lesion on the coronary angiography in accordance to European Society of Cardiology (ESC) Guidelines on

Management of Acute Myocardial Infarction in Patients Presenting with ST-Segment Elevation (2017). All patients have undergone successful angioplasty of infarct-related artery. Group 2: Patients with normal coronary angiogram diagnosed with acute myocarditis due to the clinical presentation and typical findings in CMR. EMB was performed in no cases. Diagnosis of acute myocarditis was confirmed according to the updated 2018 Lake Louise criteria in whole population [7, 21]. Only 7 symptomatic patients fulfilling the criteria of age and ST-segment elevation in ECG on admission but finally diagnosed as aortic dissection, toxin abuse, takotsubo syndrome, pericarditis or myocardial infarction type 2 were excluded from the study. In case of patients with toxin abuse or takotsubo syndrome, the CMR was also performed. In addition, patients aged <18 years and patients with poor TTE image quality were not included.

All the data were collected from patients' clinical notes and electronic records and independently assessed by two investigators.

Clinical data including typical symptoms, history of recent infection, cardiovascular risk factors—arterial hypertension, family history of CAD, smoking, diabetes mellitus, obesity defined as a body mass index (BMI) ≥30kg/m2 were reviewed. With the reference to the laboratory findings, beside C-reactive protein (CRP) and white blood cells count (WBC), we assessed the high-sensitive cardiac troponin T (TnT) and CK-MB mass level as a biomarker for the myocardial damage. The TnT cut-off point was 14 ng/l.

The study was conducted in accordance with the Declaration of Helsinki, and the protocol was approved by the Local Ethics Committee (No RNN/03/20/KE). All data were fully anonymized. The informed consent was not required.

## Echocardiography

TTE was performed on the third day of hospitalization using a high-definition Vivid E9 Ultrasound System with a 3.5-Mhz transducer (GE Healthcare, Horten, Norway) in accordance with the European Association of Cardiovascular Imaging (EACVI) recommendations. Images were optimized for gain, compression, depth and sector width and acquired at frame rates of 70–90 frames/s. Three levels (basal, middle and apical) of LV short axis and three LV apical views (4-,3- and 2-chamber) were acquired in the left lateral decubitus position during a breath hold. LV diameters were measured in long-axis parasternal standard views. LV ejection fraction (EF) and left atrial (LA) volume were assessed by means of biplane modified Simpson's rule from the apical 4- and 2-chamber views, as appropriate. LA volume was indexed to body surface area (BSA). Tricuspid annular plane systolic excursion (TAPSE) in M-mode measurement, transmitral blood flow pattern and tricuspid regurgitant velocity by means of pulsed and continuous Doppler wave respectively were also obtained. Tissue doppler imaging was used to calculate E/E' ratio and assess the diastolic function. Datasets were digitally stored for offline analysis (EchoPAC, Horten, Norway, version 201 software).

## Speckle tracking echocardiography

Echocardiographic strain imaging is a new non-invasive method for assessment of myocardial function. A 2D speckle tracking analysis was retrospectively performed using a commercially available software package (EchoPAC, Horten, Norway, version 201 software) by the two experienced and independent observers. The myocardial borders were manually traced during end systole in three apical views and in three levels of the short axis views to analyse longitudinal (myocardial shortening in the long-axis plane), radial (myocardial thickening in a short-axis plane) and circumferential (myocardial shortening in a short-axis plane) strain. The ROI (region of interest) was defined by the endocardial border. Images were automatically divided

into six standard segments in each view and the time-strain curves were generated. If the software program deemed a segment "unacceptable", a manual adjustment was performed. Peak systolic strain was determined as the maximum value of the strain during systole. Global longitudinal, radial and circumferential strain were defined as the average strain at end systole in 18 segments. A specific option of the software for three-layers analysis was used to calculate endocardial, mid-wall and epicardial GCS with relative curves. The ratio of endocardial GCS to epicardial GCS (Endo/Epi ratio) was figured for the assessment of the strain gradient [20, 22–24]. Normal ranges of reference values for layer-specific strain were based on Nagata et al. [20].

## Cardiac magnetic resonance imaging

In patients with normal coronary arteries, the CMR was done as soon as the clinical situation allowed during the same hospitalisation using the 1.5-Tesla MRI scanner (Siemens Magnetom Avanto) with phased-array body coil, ECG monitoring and enhancing contrast Gadovist, Bayer Schering Pharma, Berlin, Germany and dedicated syngo.via MR Cardiac Analysis software. Diagnosis of acute myocarditis was confirmed according to the updated 2018 Lake Louise criteria on the whole population [1, 7, 21, 25]. According to the protocols recommended by the Society for the Cardiovascular Magnetic Resonance, regional or global myocardial oedema and non-ischaemic myocardial injury were identified as areas of high-signal intensity T2-weighted imaging and the regional late gadolinium enhancement (LGE) signal increase, respectively [26].

## Statistical analysis

All the statistical analyses were performed using the Statistica Software ver. 13.1 (StatSoft Inc., Tulsa, OK, USA). The classified variables were expressed as numbers and percentages, while the continuous variables were expressed as mean ± standard deviation. The continuous values were analysed with the Mann-Whitney U test whereas the Chi-square test was used for the discrete values. A receiver-operator characteristic (ROC) curve investigation was constructed to select optimal cut-off values of strain measurements. In addition, the analysis was expanded to Spearman's rank correlation coefficient. A p values <0.05 was considered statistically significant.

## Results

69 symptomatic patients ≤45 years old including 13 females (18.8%) and 56 males (81.2%) were enrolled in this study. The mean age was 28 ± 8 and 40 ± 4 for CMR-confirmed acute myocarditis and MI respectively (p = 0.000). Demographic, clinical characteristics and laboratory parameters are presented in Table 1.

Nearly all patients had chest pain. 78.5% of patients with myocarditis reported the history of recent infection (p = 0.000). No significant differences between the groups were revealed concerning diabetes mellitus, family history of CAD, symptoms and heart rate on admission. The main cardiovascular risk factors: smoking, hypertension and obesity (defined as BMI ≥30kg/m2) were more frequent in patients finally diagnosed as MI (p<0.05). Patients with the CMR-confirmed myocarditis had significantly lower value of systolic and diastolic blood pressure measured on admission with p = 0.000 for both.

The TnT level on admission as well as after 24 hours were abnormal in groups but revealed different release patterns. Amid patients diagnosed with myocarditis there was observed higher TnT level on admission with a relatively low peak in 24-hour observation. In contrast, mean TnT level on admission for MI was lower with a higher peak after 24 hours with p<0.05.

**Table 1. Baseline characteristics.**

|  | Myocarditis | Myocardial infarction | P value |
|---|---|---|---|
|  | N = 37 | N = 32 |  |
| Age [years] | 28 ± 8 | 40 ± 4 | 0.000 |
| Sex (male) | 33 (89.2%) | 23 (71.9%) | 0.012 |
| Smoking | 16 (43.2%) | 23 (71.8%) | 0.017 |
| Hypertension | 5 (13.5%) | 15 (46.9%) | 0.005 |
| Diabetes | 2 (5.4%) | 2 (6.25%) | 0.713 |
| Obesity BMI$\geq$30 [kg/m$^2$] | 3 (8.1%) | 10 (31.2%) | 0.032 |
| Family history od CAD | 11 (29.7%) | 11 (34.4%) | 0.680 |
| Symptoms |  |  |  |
| Chest pain | 36 (97.3%) | 30 (93.8%) | 0.898 |
| Dyspnea | 3 (8.1%) | 9 (28.1%) | 0.062 |
| Recent infection * | 29 (78.4%) | 0 | 0.000 |
| Parameters on admission |  |  |  |
| HR (heart rate) [bpm] | 82 ± 13 | 83 ± 14 | 0.707 |
| Syst. blood pressure [mmHg] | 126 ± 12 | 150 ± 24 | 0.000 |
| Diast. blood pressure | 76 ± 10 | 91 ± 22 | 0.000 |
| Laboratory findings |  |  |  |
| WBC [x10$^9$ /l] ** | 10.4 ± 3.6 | 12.9 ± 6.4 | 0.108 |
| RBC [x10$^{12}$ /l] ** | 4.9 ± 0.4 | 5.0 ± 0.5 | 0.543 |
| TnT [ng/l] ** | 643 ± 626 | 641 ± 598 | 0.021 |
| TnT (after 24h) | 932 ± 778 | 2958 ± 2742 | 0.000 |
| CKMB [ng/ml] ** | 31.6 ± 32.8 | 29.8 ± 48.0 | 0.307 |
| CKMB (after 24h) | 43.2 ± 50.2 | 140.3 ± 167.0 | 0.005 |
| CRP [mg/l] (max) | 71.7 ± 67.4 | 38.8 ± 54.5 | 0.006 |

All values presented as mean ± SD or N (%).

* <3 weeks before admission.

** on admission.

In MI group the most numerous were patients with anterolateral myocardial infarction (N = 10). 7 patients presented MI of anteroseptal wall. 3 patients were diagnosed as MI of inferior, inferobasal wall and right ventricle (RV). 5 patients had isolated inferior wall infarct, 2 patients presented inferolateral MI, 4 inferior and inferobasal. Only 1 patient was diagnosed as MI of inferior wall and RV.

As far as echocardiographic features are concerned LVEF was lower in patients with myocardial infarction rather than acute myocarditis (mean 50% ± 8 vs 56% ± 7, p = 0.002). The median of impaired segments in MI and myocarditis group was 4 (IQR 3–6.5) and 1 (IQR 0–4) respectively. Diastolic posterior wall and intraventricular septum thickness were significantly higher in myocardial infarction group with p<0.05 for both. Further diastolic dysfunction measured by E/E' average was more deteriorated in patients with infarction compared to acute myocarditis (mean 8 ± 2 vs 7 ± 2, p = 0.000). There were no significant differences in left atrium volume index (LAVi), LV diastolic diameters and right ventricular function measured by TAPSE between the groups. A complete description of echocardiographic parameters is reported in Table 2.

On speckle tracking analysis, an average GLS, GRS as well as GCS including three—layers observation were significantly lower (absolute value) in patients with acute myocardial infarction (AMI) versus acute myocarditis with p<0.05. There was no significant difference in

**Table 2. Baseline echocardiographic parameters.**

|  | Myocarditis | Myocardial infarction | P value |
|---|---|---|---|
|  | N = 37 | N = 32 |  |
| PWd [mm] | 10 ± 1 | 11± 2 | 0.002 |
| IVSd [mm] | 10 ± 1 | 12 ± 2 | 0.000 |
| LVEDD [mm] | 50 ± 4 | 50 ± 5 | 0.538 |
| LVESD [mm] | 34 ± 6 | 34 ± 6 | 0.976 |
| LAVi [ml/m2] | 30 ± 5 | 30 ± 12 | 0.708 |
| TAPSE [mm] | 23 ± 3 | 22 ± 4 | 0.182 |
| LV EF [%] | 56 ± 7 | 50 ± 8 | 0.002 |
| E/A ratio | 1.5 ± 0.3 | 1.2 ± 0.3 | 0.000 |
| E/E' average | 7 ± 2 | 8 ± 2 | 0.000 |

All values presented as mean ± SD.

Endo/Epi ratio (mean 2.3 ± 0.7 vs 2.5 ± 0.6, p = 0.144). Distribution of Endo/Epi ratio values having regard to normal reference according to Nagata at al. is presented in Fig 1. Complete STE parameters are reported in Table 3.

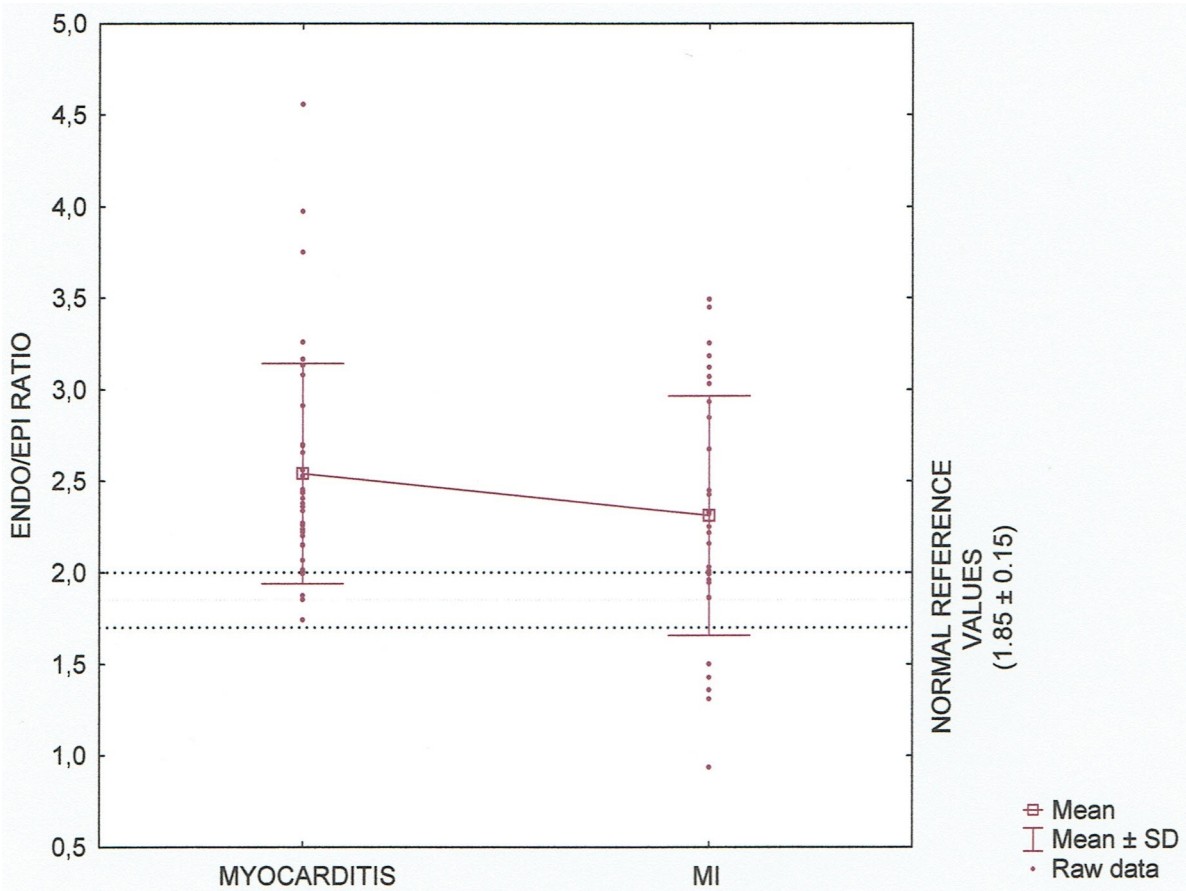

**Fig 1. Layer–specific circumferential strain (Endo/Epi ratio) in the myocarditis and MI groups.** Data are shown as Tukey boxplot and raw data, taking into account normal references values for age, p = 0.144.

**Table 3. STE data.**

|  | Myocarditis | Myocardial infarction | P value |
|---|---|---|---|
|  | N = 37 | N = 32 |  |
| **Global longitudinal strain [%]** |  |  |  |
| Apical four chamber | -18.0 ± 2.9 | -15.0 ± 4.1 | 0.001 |
| Apical two chamber | -19.2 ± 3.4 | -15.9 ± 4.2 | 0.002 |
| Apical three chamber | -18.8 ± 2.9 | -14.2 ± 7.3 | 0.000 |
| Average | -18.7 ± 2.7 | -15.4 ± 3.9 | 0.000 |
| **Global radial strain [%]** |  |  |  |
| Average | 33.4 ± 12.0 | 23.3 ± 11.5 | 0.001 |
| **Global circumferential strain [%]** |  |  |  |
| Epicardial | -10.4 ± 3.1 | -8.8 ± 3.3 | 0.036 |
| Mid-wall | -17.3 ± 3.2 | -13.8 ± 4.6 | 0.005 |
| Endocardial | -25.3 ± 6.3 | -21.0 ± 7.2 | 0.009 |
| Endo/Epi ratio | 2.5 ± 0.6 | 2.3 ± 0.7 | 0.144 |
| Average | -17.5 ± 4.3 | -14.7 ± 4.8 | 0.036 |

All values presented as mean ± SD.

An average LV GLS < (-17.5) represented the optimal cut-off value for the acute myocarditis diagnosis with the area under the ROC curve (AUC) 0.737 [95% CI 0.617–0.857] (Fig 2A). Despite preserved ejection fraction in both groups GLS was significantly lower (absolute value) in patients diagnosed as acute myocardial infarction compared to the myocarditis group. GLS in patients with myocarditis is located in the majority within normal reference values of strain correctly to age (Fig 2B). In addition, by bivariate correlation analysis, there was a strong parallel between GLS average value and LVEF in whole population (Fig 3A) and in both

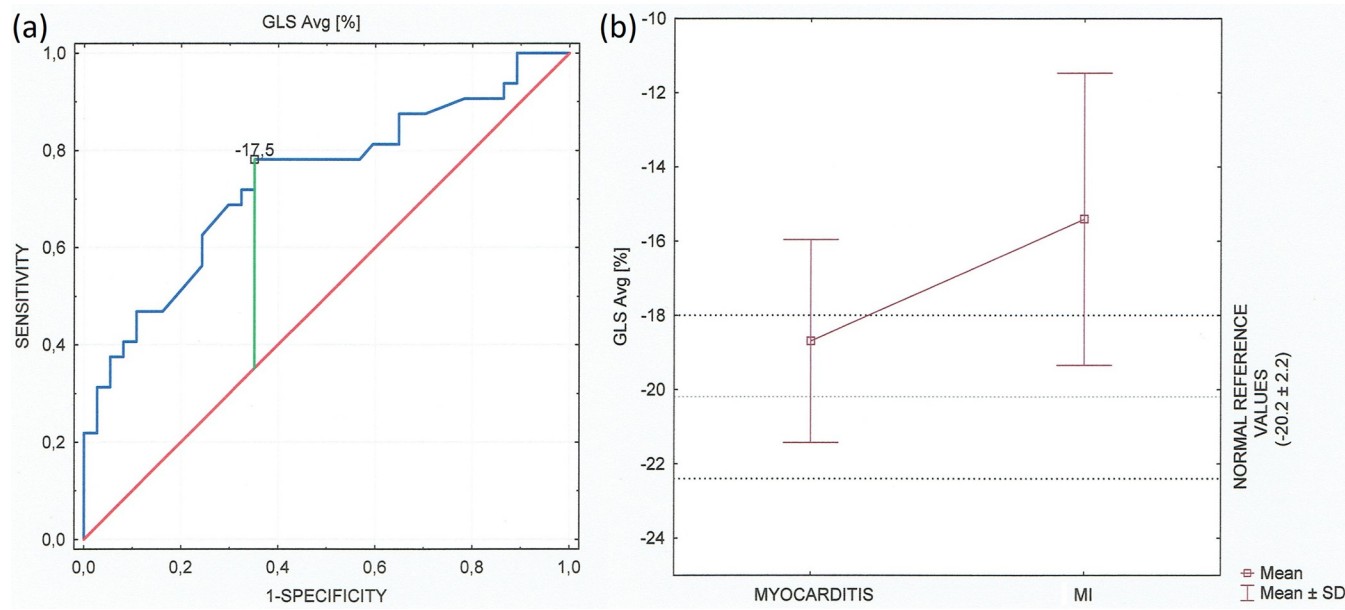

**Fig 2.** (a) ROC curve of the GLS average [%] for the diagnosis of clinically suspected myocarditis. The AUC was 0.737 (standard error = 0.061, 95% confidence interval 0.795–0.954). It shows that the sensitivity and specificity of the GLS is adequate in statistics (b) Global longitudinal strain [%] in the myocarditis and MI groups including normal reference of strain. Data are shown as Tukey boxplot, (p = 0.000).

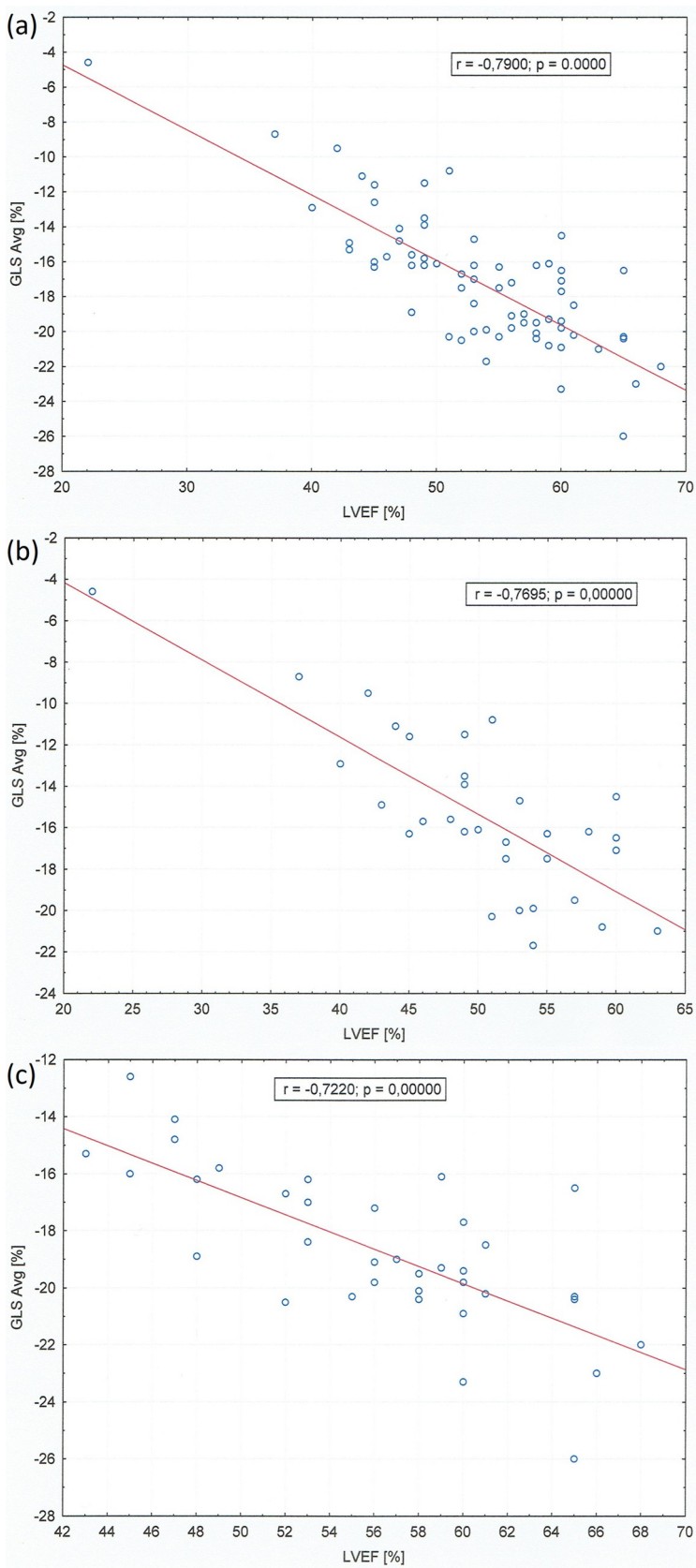

**Fig 3.** Scatter plot of a strong correlation between GLS average [%] and left ventricular ejection fraction [%] in whole population (a), myocarditis group (b) and myocardial infarction group (c).

groups separately (Fig 3B and 3C). The examples of the two different longitudinal strain bull's eye plots in patients admitted with an initial diagnosis of STEMI and with the final diagnosis of myocardial infarction and acute myocarditis are presented in Fig 4A and 4B respectively. In contrast to patients with myocarditis it confirms the ischaemic pattern of myocardial injury related strictly to heart vascularization in patients diagnosed as AMI.

The average GCS and layer–specific GCS among the groups presented as Tukey boxplots (mean ± SD) are provided in Figs 5 and 6, respectively. We also compared the results to the normal reference values of strain.

Our analysis revealed the statistically significant differences between the groups in all three layers and average (p<0.05). It also confirmed the discriminative pattern of myocardial injury with three—layers damage in patients with ischaemic myocardial infarction and midwall to epicardial distribution excluding endocardial location in patients diagnosed as acute myocarditis.

## Discussion

Our study suggests that a combination of clinical and laboratory findings in conjunction with echocardiography parameters with particular reference to STE of LV might become a valuable tool for confirming the diagnosis of acute myocarditis in patients with chest pain, presenting ST-segment elevation on admission and normal coronary arteries in coronary angiography. Nowadays, EMB is the only method enabling a certain diagnosis of myocarditis, however infrequently followed in clinical practice [1, 6, 7, 12]. Cardiac magnetic resonance is currently the preferred non-invasive tool for myocardial tissue characterization providing the essential information about morphology and function of cardiac structures, myocardial oedema, hyperemia and myocardial scarring in LGE [1, 7]. In myocardial scar the wash out of gadolinium contradistinctively to normal myocardium is deferred. Different patterns of LGE indicate the ischaemic or non-ischaemic cause of myocardial injury. Ischaemic necrosis expands from the

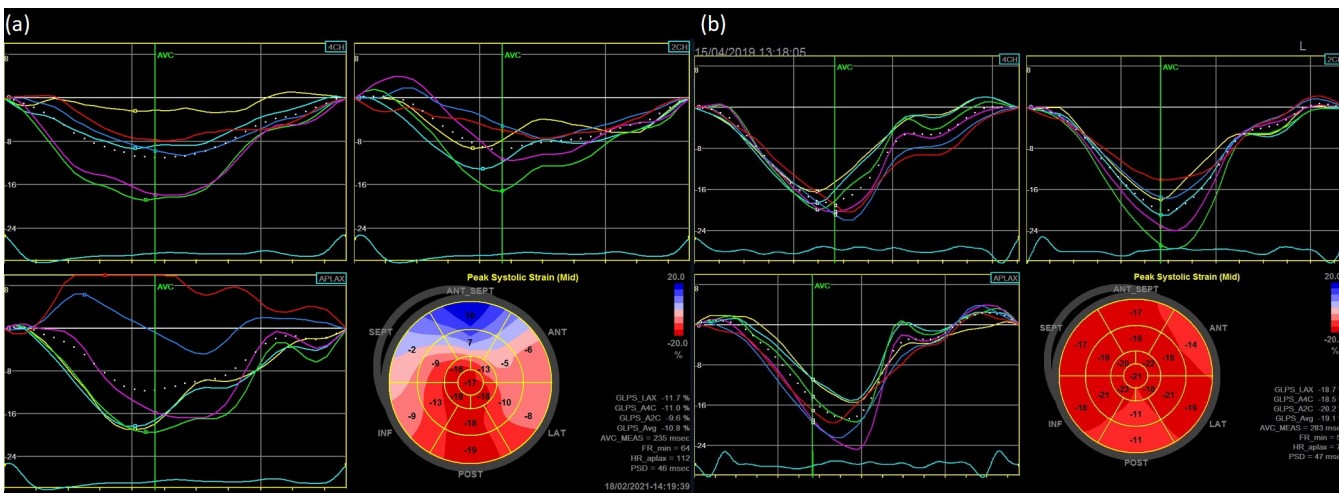

**Fig 4.** Example of the longitudinal strain bull's eye plot derived from two-dimensional speckle tracking imaging in patients admitted with an initial diagnosis of anterolateral STEMI and with the final diagnosis of myocardial infarction resulted from occlusion of the left anterior descending artery (a) and acute myocarditis (b).

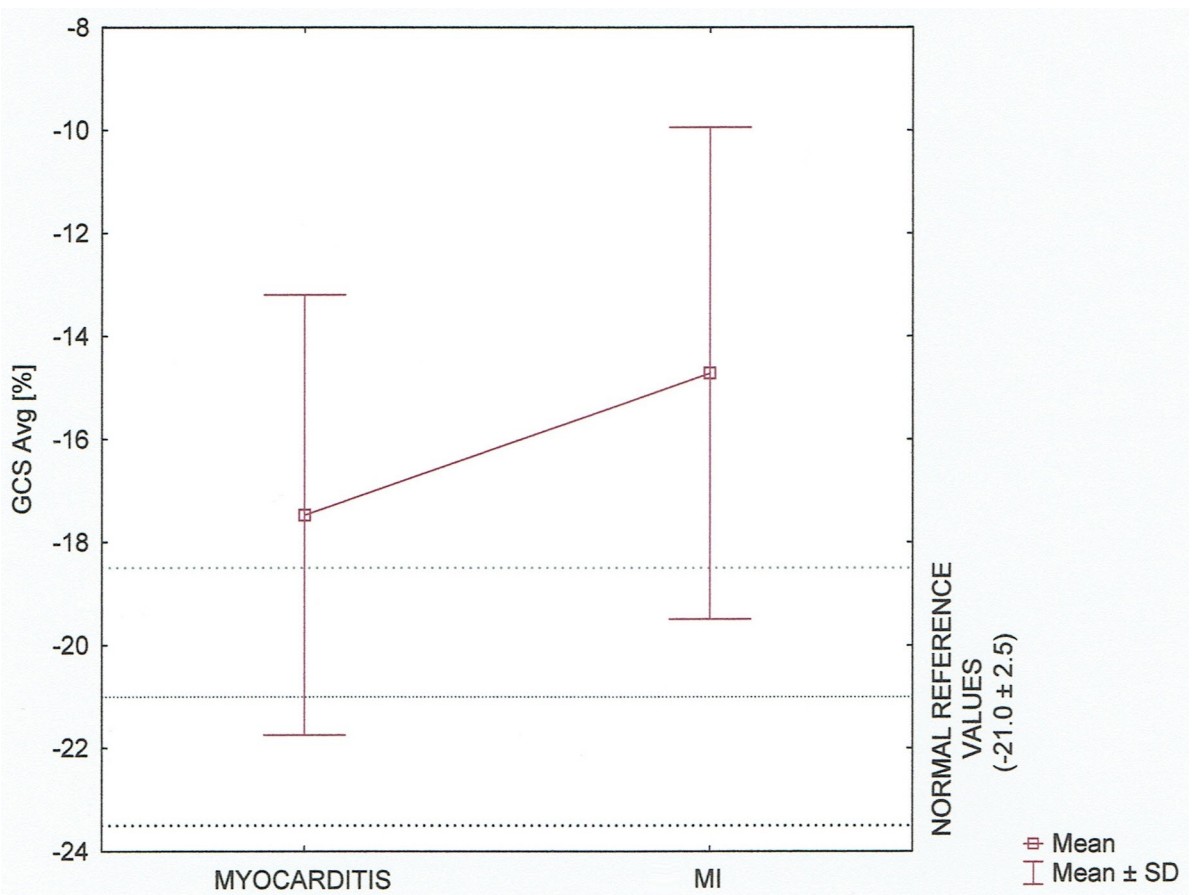

**Fig 5. Global circumferential strain (average) [%] concerning normal reference values.** Tukey boxplot (mean ± SD).

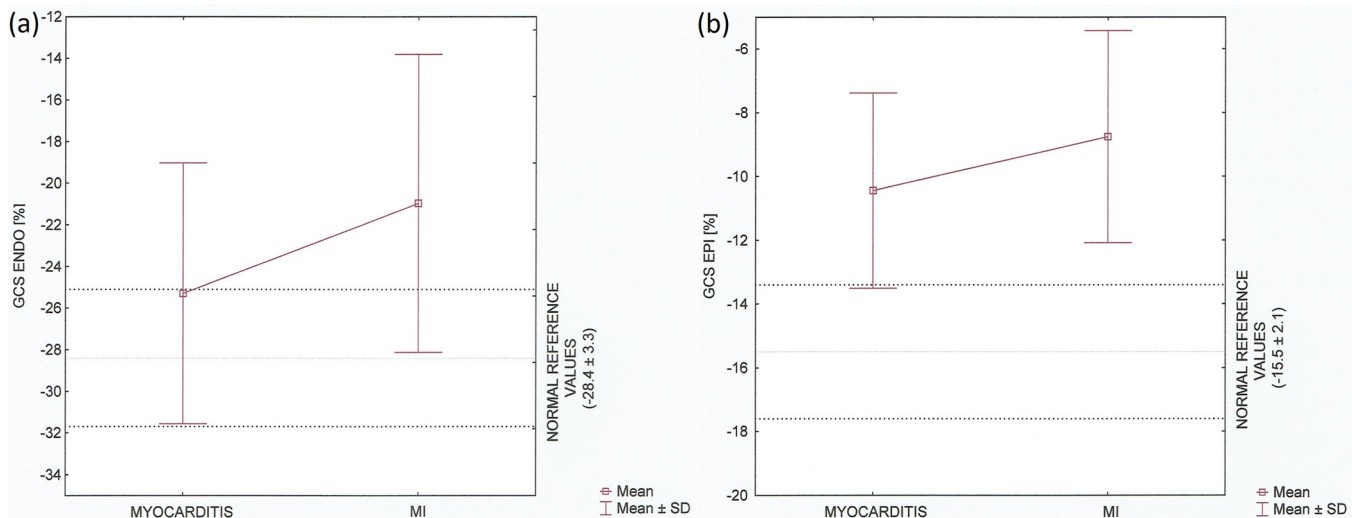

**Fig 6. Global circumferential endocardial and epicardial strain in the myocarditis and MI groups.** Data are shown as Tukey boxplots, taking into account normal reference values for age, p<0.05 (a) Endocardial GCS [%]. (b) Epicardial GCS [%].

endocardium to epicardium revealed the transmural character of changes, whereas in non-ischaemic necrosis LGE is limited to midwall and epicardial layers [17, 18, 27–29]. Although basic TTE, including LVEF assessment, is the most often used tool to evaluate LV function, it is often not sufficient enough to detect subtle myocardial dysfunction [30]. Novel echocardiographic techniques such as STE constitute a great advantage to provide a sensitive quick diagnostic examination in the acute phase of disease [7, 12]. In our study, we found the corresponding to CMR LGE images layer–specific strain impairment including transmural distribution in patients diagnosed as acute coronary syndrome and non–transmural, subepicardial involvement in patients with acute myocarditis. In the latter we observed the values of endocardial GCS within the normal ranges for age in contrast to patients with confirmed obstructive CAD, p = 0.009. The presence of mainly epicardial strain impairment in combination with preserved endocardial strain value can be supportive to exclude ischemic heart disease in case of acute coronary syndrome–like presentation of myocarditis especially when CMR is not easily available [6, 27]. Moreover, according to the references, it seems that global longitudinal myocardial function is reduced when extensive transmural myocardial damage is observed [18]. In our research, we confirmed the statistically significant difference (p = 0.000) in GLS values between the myocarditis and MI group with higher values (absolute values) in patients diagnosed as AM and the cut-off point for suspected myocarditis defined as the GLS < (-17.5) (AUC 0.737, 95% CI 0.795–0.954). On the other hand, there was no significant difference as far as Endo/Epi ratio is concerned. It might be associated with the observed immediate improvement of endocardial strain in patients with acute STEMI undergoing primary percutaneous coronary intervention (PCI) [31]. Myocardial injury due to ischemic heart disease spreads from the endocardium towards epicardium [27]. According to Atıcı et al. in patients with the phenomenon of no-reflow after PCI multilayer strain imaging revealed lower strain values particularly in endocardial layer [32]. Non-invasive diagnostics including speckle tracking echocardiography may constitute a new starting point for consideration about separation the myocarditis group without additional imaging in patients with an initial diagnosis of STEMI on admission at which the obstructive CAD had been excluded.

## Study limitations

Some limitations of this study need to be noted. Initially, this is a single-center, retrospective study resulting in the relatively small number of patients. Secondly, despite the fact that, according to the position of experts group in ESC guidelines, EMB is still regarded as a gold standard for the diagnosis of myocarditis, it was not performed in any case. In our study, the diagnosis of acute myocarditis was confirmed in compliance with the updated 2018 Lake Louise criteria in the whole population (increased signal intensity in T2-weighted imaging and increased signal intensity with a non-ischeamic distribution pattern in LGE images).

There are few studies providing the information about the normal ranges of the left ventricular strain in adults [19, 33, 34]. For further optimization multi-center studies are required.

## Conclusion

In patients with AMI a significant reduction of global and three-layers strains compared to patients with myocarditis was detected. Furthermore, corresponding to CMR LGE images layer–specific strain impairment might provide valuable information to complement the diagnosis of acute myocarditis especially when CMR is not easily available in patients without coronary artery disease.

## Supporting information

**S1 Table. Minimal data set.**
(PDF)

## Author Contributions

**Conceptualization:** Paulina Wieczorkiewicz, Karolina Supel, Marzenna Zielinska.

**Data curation:** Paulina Wieczorkiewicz, Katarzyna Przybylak.

**Formal analysis:** Paulina Wieczorkiewicz, Karolina Supel, Katarzyna Przybylak, Michal Kacprzak, Marzenna Zielinska.

**Investigation:** Paulina Wieczorkiewicz, Karolina Supel, Katarzyna Przybylak, Michal Kacprzak, Marzenna Zielinska.

**Methodology:** Paulina Wieczorkiewicz, Karolina Supel, Michal Kacprzak.

**Resources:** Paulina Wieczorkiewicz, Katarzyna Przybylak.

**Software:** Katarzyna Przybylak, Michal Kacprzak, Marzenna Zielinska.

**Supervision:** Marzenna Zielinska.

**Validation:** Marzenna Zielinska.

**Visualization:** Paulina Wieczorkiewicz, Marzenna Zielinska.

**Writing – original draft:** Paulina Wieczorkiewicz.

**Writing – review & editing:** Karolina Supel, Katarzyna Przybylak, Michal Kacprzak, Marzenna Zielinska.

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
