## [Decision Letter · Decision Letter 0]

1 May 2022

PONE-D-22-08288ACUTE CORONARY SYNDROME VERSUS ACUTE MYOCARDITIS IN YOUNG ADULTS – VALUE OF SPECKLE TRACKING ECHOCARDIOGRAPHYPLOS ONE

Dear Dr. Wieczorkiewicz,

Thank you for submitting your manuscript to PLOS ONE. After careful consideration, we feel that it has merit but does not fully meet PLOS ONE’s publication criteria as it currently stands. Therefore, we invite you to submit a revised version of the manuscript that addresses the points raised during the review process. Specifically, the statistics should be implemented before consideration of further revisions. 

We look forward to receiving your revised manuscript.

Kind regards,

Antonio Cannatà

Academic Editor

PLOS ONE

Journal Requirements:

Additional Editor Comments:

In addition to reviewers' comments, the authors should provide statistical correction for potential confounders especially when validating the role of GLS. Although may suffer collinearity, the variable might be tested against LVEF to highlight its predictive role.

Reviewers' comments:

Reviewer's Responses to Questions

**Comments to the Author**

1. Is the manuscript technically sound, and do the data support the conclusions?

Reviewer #1: No

Reviewer #2: Partly

2. Has the statistical analysis been performed appropriately and rigorously? 

Reviewer #1: No

Reviewer #2: I Don't Know

3. Have the authors made all data underlying the findings in their manuscript fully available?

Reviewer #1: Yes

Reviewer #2: Yes

4. Is the manuscript presented in an intelligible fashion and written in standard English?

Reviewer #1: Yes

Reviewer #2: Yes

5. Review Comments to the Author

Reviewer #1: In this small retrospective study, authors aim at investigating the value of speckle tracking echocardiography in the differential diagnosis between acute coronary syndrome and acute myocarditis. Due to the possible clinical overlap between these two conditions the differential diagnosis is challenging, sometimes impossible in the acute phase, therefore coronary angiography is still largely performed in the emergency setting to exclude SCA. For this reason, the topic of the paper is very actual, however there are several points pf the work which should be addressed.

Major comments:

- From my point of view, the main limitation of the paper is the lack of clinical implications of the study. Authors conclude that patients with AMI have a significant reduction of global and three layers strains as compared to those with myocarditis. However, it seems difficult to consider using echocardiography strain analysis to discriminate between patients who should undergo coronary angiography and those who should not in the setting of acute chest pain and ST segment elevation ant EKG.

- In the paragraph “conclusion” authors state that “specific strain impairment might provide valuable information to complement the diagnosis of acute myocarditis when CMR is not easily available”. I would rather say that “specific strain impairment might provide valuable information to complement the diagnosis of acute myocarditis when CMR is not easily available in patients without coronary artery disease”.

- Due to the retrospective nature of the study a multivariate analysis should be performs in order to correct the results for confounders especially considering that patients with myocardial infarction were older, had higher Tn levels and more cardiovascular risk factors.

Minor comments:

- Page 3 authors say: “EMB is still the diagnostic gold standard but extremely limited in the clinical practice. Please consider citing the recently published Position Statement by Seferovic et al (DOI: 10.1016/j.cardfail.2021.04.010);

- Please check minor English typos.

Reviewer #2: In the present paper the authors investigate the diagnostic value of clinical and instrumental tools, focusing in particular on speckle tracking echocardiography, in patients with acute coronary syndrome-like myocarditis and acute myocardial infarction. They found that patients with AMI have a significant reduction of global strains and a different layer involvement compared to patients with myocarditis. Thus, they suggest that speckle tracking analysis may become a valuable tool to differentiate these two entities.

The topic is very actual and interesting, however some issues should be addressed:

- As the authors have mentioned, the paper presents some limitations like the small sample size or the absence of a histological confirmation of the myocarditis diagnosis. Furthermore, for patients with AMI diagnosis some information is lacking like the STEMI localization and if there is a different a bigger strain impairment in patients with a more estensive infarction. Also, it is not specify if all AMI patients have undergone PCI.

- There are conflicting data about the differences in layer-specific strain impairment. The major finding of the present study is that the presence of a mainly epicardial strain impairment with preserved endocardial strain can suggest a diagnosis of myocarditis and exclude ACS, in which the three layers are involved. Conversely, the value of endocardial strain, which should be the most compromised, has the higher value (-21) and there was not a significant difference in endo/epi ratio between the two groups.

- The fact that the absolute GLS value was significantly lower in patients diagnosed as AMI compared to the myocarditis group does not provide a relevant additional information given that also the LVEF is lower in the AMI group.

- It is curious that PWd and IVSd were significantly higher in myocardial infarction group because, despite the higher prevalence of hypertension in this group, in the myocarditis group it is expected a ventricular pseudohypertrophy due to myocardial edema.

- It could be interesting to complete the speckle tracking analysis with also the right ventricolare values (in particular for inferior STEMI).

6. PLOS authors have the option to publish the peer review history of their article (what does this mean?). If published, this will include your full peer review and any attached files.

Reviewer #1: No

Reviewer #2: No

---

## [Author Response · Author response to Decision Letter 0]

31 May 2022

Dear Sir/Madam,

I would like to submit the revised version of the manuscript entitled “Acute coronary syndrome versus acute myocarditis in young adults – value of speckle tracking echocardiography” by Wieczorkiewicz P, Supel K, Przybylak K, Kacprzak M, Zielinska M to be considered for publication as an original article in the PLOS ONE. 

On behalf of all co-authors, I would like to thank you very much for all the reviewers’ comments and suggestions. 

Appreciating their input and effort, we have revised the manuscript in accordance with all the recommendations. 

The first and foremost we have supplemented the statistics with the scatter plots of a strong correlation between GLS average and left ventricular ejection fraction in whole population and in both groups separately, what confirms the predictive role of GLS in whole analysis. The similar pattern of the correlations proves that this relation does not differentiate the groups. 

Other changes are described in the Response to the Reviewers. 

We believe that these corrections will improve the manuscript in terms of its educational value. 

We are looking forward to hearing from you at your earliest convenience. 

Best regards, 

Paulina Wieczorkiewicz

---

## [Decision Letter · Decision Letter 1]

5 Jul 2022

ACUTE CORONARY SYNDROME VERSUS ACUTE MYOCARDITIS IN YOUNG ADULTS - VALUE OF SPECKLE TRACKING ECHOCARDIOGRAPHY

PONE-D-22-08288R1

Dear Dr. Wieczorkiewicz,

We’re pleased to inform you that your manuscript has been judged scientifically suitable for publication and will be formally accepted for publication once it meets all outstanding technical requirements.

Kind regards,

Antonio Cannatà

Academic Editor

PLOS ONE

Additional Editor Comments (optional):

Reviewers' comments:

Reviewer's Responses to Questions

**Comments to the Author**

1. If the authors have adequately addressed your comments raised in a previous round of review and you feel that this manuscript is now acceptable for publication, you may indicate that here to bypass the “Comments to the Author” section, enter your conflict of interest statement in the “Confidential to Editor” section, and submit your "Accept" recommendation.

Reviewer #3: All comments have been addressed

2. Is the manuscript technically sound, and do the data support the conclusions?

Reviewer #3: Yes

3. Has the statistical analysis been performed appropriately and rigorously? 

Reviewer #3: Yes

4. Have the authors made all data underlying the findings in their manuscript fully available?

Reviewer #3: Yes

5. Is the manuscript presented in an intelligible fashion and written in standard English?

Reviewer #3: Yes

6. Review Comments to the Author

Reviewer #3: no additive comment

the manuscript is OK

the results and the discussions are OK

the figure are OK as well as the references

7. PLOS authors have the option to publish the peer review history of their article (what does this mean?). If published, this will include your full peer review and any attached files.

Reviewer #3: No

---

## [Editor Report · Acceptance letter]

29 Jul 2022

PONE-D-22-08288R1 

Acute Coronary Syndrome versus Acute Myocarditis in young adults – value of Speckle Tracking Echocardiography 

Dear Dr. Wieczorkiewicz:

I'm pleased to inform you that your manuscript has been deemed suitable for publication in PLOS ONE. Congratulations! Your manuscript is now with our production department. 

Kind regards, 

on behalf of

Dr. Antonio Cannatà 

Academic Editor

PLOS ONE